# The Molecular, Morphological and Genetic Characterization of Glyphosate Resistance in *Conyza bonariensis* from South Africa

**DOI:** 10.3390/plants11212830

**Published:** 2022-10-24

**Authors:** Martha N. Okumu, Petrus J. Robbertse, Barend J. Vorster, Carl F. Reinhardt

**Affiliations:** 1Department of Plant and Soil Sciences, University of Pretoria, Private Bag X20, Hatfield 0028, South Africa; 2Faculty of Natural and Agricultural Sciences, Forestry and Agricultural Biotechnology Institute, University of Pretoria, Private Bag X20, Hatfield 0028, South Africa

**Keywords:** herbicide resistance, *C. bonariensis*, dose–response, leaf morphology, EPSP synthase, heredity of resistance

## Abstract

Six *Conyza bonariensis* (L.) Cronquist populations were screened in a pot experiment at the University of Pretoria’s Hatfield experimental farm to evaluate and confirm the degree of glyphosate response. Resistance factors ranged from 2.7- to 24.8-fold compared to the most susceptible biotype. Partial sequencing of the *5-enolpyruvylshikimate-3-phosphate synthase (EPSPS)* gene found no mutation at the Thr102, Ala103 or Pro106 positions. *EPSPS* mRNA expression levels in glyphosate-resistant biotypes (Swellendam and Piketberg seed sampling sites) were comparable or lower than those in susceptible biotypes (George and Fauresmith sites). Additionally, the highest expression level was reported in the susceptible Fauresmith biotype. These results indicate that glyphosate resistance in the tested resistant biotypes is not caused by target-site mutations and *EPSPS* gene amplification. Leaf surface characteristics can influence the spread and subsequent absorption of glyphosate. The study established non-significant results in the amount of leaf wax and insufficient mean separations in cuticle thickness and trichome density data. Therefore, the observed differences in response of biotypes to glyphosate treatment could not be attributed conclusively to differences in the leaf morphological characteristics investigated. Results from the inheritance study were consistent with glyphosate resistance being inherited in an incompletely dominant manner when plants were treated with glyphosate herbicide at 900 g ae ha^−1^.

## 1. Introduction 

Glyphosate, whose mechanism of action is the exclusive inhibition of 5-enolpyruvylshikimate-3-phosphate synthase (EPSPS; EC 2.5.1.19), is a highly effective herbicide with a relatively small economic and environmental footprint globally [1,2]. Glyphosate has high water solubility, penetrates the leaf cuticle relatively easily and is translocated symplastically in the phloem from assimilate sources to sinks such as apical meristems [3]. Globally, the high use frequency of glyphosate in cropping systems has given rise to glyphosate-resistant weeds [2,4]. Hairy fleabane (*C. bonariensis*) is one of the most difficult-to-control and glyphosate-resistant weed species, particularly in conservation tillage cropping systems in both the summer and winter rainfall regions of South Africa.

Herbicide resistance in weed species has been categorized into target- and nontarget-site mechanisms [4,5]. Target-site herbicide resistance in glyphosate-resistant weeds is a result of mutations in *EPSPS* gene leading to changes at the following amino acid positions, threonine (Thr102), alanine (Ala103) and proline (Pro106) [6]. These mutations prevent the herbicide from binding to the target site, thereby reducing the efficacy of glyphosate’s action on the EPSPS enzyme [6,7]. Enrichment of these mutations in weed populations gives rise to the development of glyphosate resistance [8]. *EPSPS* gene mutations have been reported in a number of weeds, including horseweed (*Conyza canadensis* L.) [9], goosegrass (*Eleusine indica* L.) [10,11,12], junglerice (*Echinochloa colona* L.) [13], tall waterhemp (*Amaranthus tuberculatus* Moq. Sauer) [14], and hairy beggarticks (*Bidens pilosa* L.) [15], among others [16]. Similarly, mutations can occur in the DNA sequence of *EPSPS* gene, resulting in an increased expression of glyphosate’s target protein [17]. *EPSPS* overexpression is the result of an increase in gene transcript levels or number of gene copies [7,18]. Identified mechanisms of glyphosate resistance in *C. bonariensis* include impaired translocation, most likely due to vacuolar sequestration and increased production of *EPSPS* transcripts [19,20].

In the non-target-site resistance mechanism(s), there is a reduction in the amount and rate of herbicide accumulation at the target site [5]. Non-target site resistance can be due to decreased herbicide penetration into the plant, differential or reduced uptake and/or translocation, increased rate of sequestration or metabolism of the herbicide molecule. 

Vacuolar sequestration has been linked to ABC transporters, with an overexpression of, in particular the *M10* and *M11* genes increasing the survival of weeds after glyphosate treatment [21]. Differential herbicide uptake could be due to morphological characteristics of the plant, for example, hairiness or over-production of epicuticular waxes [22]. 

Leaf morphological characteristics including the cuticle, epicuticular waxes and trichomes influence surface tension and penetration of the herbicide [22]. The penetration and subsequent absorption of foliar-applied herbicides is influenced by the thickness of the cuticle, with thicker cuticles presenting more resistance compared to thinner cuticles [23]. Reduced foliar uptake and spray retention was found in resistant compared to susceptible plants in Italian ryegrass (*Lolium multiflorum* Lam.) and perennial ryegrass (*Lolium perenne* L.) [24,25]. These observations were attributed to variability in cuticular properties, with the resistant biotype having thicker cuticles [24].

The trichome density affects the coverage of foliar-applied herbicides in that spray droplets falling on large trichomes may break or bounce off the surface of the leaf [26]. Additionally, small-sized spray droplets lodge between the hairs (trichomes), thus reducing the volume reaching the epidermal surface [27]. The hydrophobic nature of plant waxes decreases wettability of leaf surfaces, and therefore act as barriers to water-soluble foliar-applied herbicides [28]. 

The herbicide resistance inheritance pattern has been reported to determine the rate at which selection within a population will increase the enrichment of the resistance gene. Glyphosate resistance has been determined to be inherited as an incompletely dominant trait in *C. canadensis* [29,30], rigid ryegrass (*Lolium rigidum* Gaud.) [31,32] and *E. indica* [33]. In *C. bonariensis*, Okada and Jasieniuk [34] reported both single-locus and additive two-loci inheritance patterns in the populations they studied. 

Previous studies on *C. bonariensis* whose seeds were sampled from western and southern Cape regions of South Africa showed glyphosate resistance factors (RF) of up to 26.9-fold in the most resistant biotype compared to the susceptible biotype [35]. Mechanisms of glyphosate resistance in South African biotypes of *C. bonariensis* have not been reported. This study extends previous work by investigating the molecular and morphological aspects of glyphosate resistance, as well as the mode of inheritance. 

Shikimate acid levels in the shikimate assay in a preliminary study showed significantly lower levels in glyphosate-resistant (GR) compared to glyphosate-susceptible (GS) biotypes [35]. This means that the shikimate pathway is inhibited and required a molecular investigation on *EPSPS* gene by exploring the target-site mechanisms of resistance. The hypothesis tested was that a target mechanism of resistance; either a point mutation(s) or overexpression of the target site is responsible for the previously observed differences in the response of the biotypes to glyphosate treatment. It was hypothesized that differences in response to glyphosate treatment among the biotypes are attributed to differences in leaf morphological characteristics (cuticle thickness, amount of wax and trichome density). The present study used transmission electron microscopy (TEM) to determine the cuticle thickness. To our knowledge, no research on cuticle thickness has been carried out on *C. bonariensis* using this technique. It was therefore considered important to carry out a study to examine the morphological characteristics as a possible explanation for differential response of *C. bonariensis* biotypes to glyphosate treatment. Based on previous studies on inheritance of glyphosate resistance in various weed species, including *C. bonariensis*, we hypothesize an incompletely dominant inheritance pattern. 

The aims of this study were to: (i) confirm the glyphosate-response status of six *C. bonariensis* populations by use of a dose–response approach, (ii) elucidate the molecular mechanism(s) of glyphosate resistance by using *EPSPS* gene sequencing and expression analyses, (iii) describe the morphological characterization of highly resistant and susceptible populations, and (iv) identify the mode of inheritance of the glyphosate resistance traits in *C. bonariensis* biotypes from South Africa. 

## 2. Results

### 2.1. Dose–Response Experiment of 6 Biotypes 

Phytotoxic effects were observed on the leaves of all biotypes five to seven days after treatment (DAT) with the young developing leaves turning yellow (chlorotic). In susceptible biotypes (George and Fauresmith), there was wilting, and the mild chlorosis advanced into necrotic lesions that spread on all leaves, followed by browning and death by 21 days after treatment (DAT). In the resistant biotypes, the mild chlorosis was replaced with green new growth at the center of the rosettes. The glyphosate-resistant plants recovered from the herbicidal effect and showed robust growth. 

Resistance factors or RF were calculated by obtaining the ratio between the GR_50_ of resistant (GR) biotype and the GR_50_ of the most susceptible (GS) biotype from George. The GS biotype from Fauresmith showed a resistance factor of 2.7-fold and only required half the normal-use rate for 50% reduction in fresh weight (Table 1). The GR biotypes displayed resistance factors ranging from 4.7- to 24.8-fold with GR_50_ values varying from 768.82 to 4014.79 g ae ha^−1^. The GR biotype from Piketberg demonstrated a high resistance factor of 24.8-fold compared to the GS biotype from George and needed approximately 4.5 times the normal-use rate for reduction of half its fresh weight. The GS biotypes were collected from a hops (*Humulus lupulus* L.) orchard (George) and alfalfa (*Medicago sativa* L.) field (Fauresmith), which both have no history of glyphosate use, while the GR biotypes were from vineyards (*Vitis vinifera* L.) (Durbanville, Piketberg and Worcester) and a wheat (*Triticum aestivum* L.) field (Swellendam) with a history of regular glyphosate use. Using the estimated GR_50_ values, dose–response curves were plotted to graphically display the response of biotypes to glyphosate (Figure 1). 

### 2.2. Molecular Characterization of EPSPS Gene

#### 2.2.1. Partial Sequencing of *EPSPS* cDNA 

A 492 bp region of the *EPSP synthase* gene coding the region of proline 106, where mutations conferring glyphosate resistance have previously been reported was sequenced from cDNA. Alignment of the sequences from the five *C. bonariensis* biotypes did not show any point mutations coding residues of amino acids threonine 102, alanine 103 or proline 106, nor any amino acid changes that were unique to resistant biotypes (Figure 2). 

#### 2.2.2. *EPSPS* cDNA Expression Level

To determine the possibility of the mechanism of glyphosate resistance in the tested biotypes being an overexpression of *EPSPS* gene, quantitative RT-PCR was carried out on the cDNA, relative to *Actin* gene. *EPSPS* mRNA expression levels were determined two times after glyphosate treatment (4 DAT and 6 DAT). The level of expression was up-regulated at 6 DAT in George (GS), Fauresmith (GS) and Swellendam (GR) biotypes. Expression levels were 1.24-, 2.24- and 1.19-fold compared to the untreated control, respectively (Figure 3). Although Piketberg was considered resistant according to the glyphosate screening study, the expression level was 0.85-fold, which was significantly lower than susceptible biotypes. The expression level in the susceptible Fauresmith biotype gradually increased over time, with the highest expression of 2.24-fold being recorded at 6 DAT compared to the untreated control.

### 2.3. Leaf Morphological Characterization 

#### 2.3.1. Cuticle Thickness Examination 

The cuticle thickness was significantly greater in GR Swellendam biotype compared to both the GS Fauresmith and GS George biotypes, but not significantly different from the GR Piketberg biotype. However, the cuticle thickness tended to be higher in both the GR as compared to both the GS biotypes (Table 2 and Appendix A). The average cuticular membrane thickness ranged between 0.11 µm in Swellendam (GR) and 0.07 µm in George (GS) biotypes (Table 2). Ultrastructure images of cuticles of all biotypes were highly corrugated or folded (Appendix A).

#### 2.3.2. Trichome Density Determination using Light Microscopy and Scanning Electron Microscopy

*C. bonariensis* trichomes are simple, non-glandular trichomes with micropapillate surface sculpturing (Appendix A). Adaxial and abaxial leaf surfaces of *C. bonariensis* were examined by light microscopy and the number of trichomes per mm^2^ counted. The results from analysis of variance showed significant differences in the leaf surface (*p* < 0.0001), with more trichomes being found on the adaxial than on the abaxial leaf surfaces (Table 3 and Appendix A). The mean number of trichomes on the adaxial leaf surface was 6.52 per mm^2^, compared to 5.79 per mm^2^ on abaxial surface (Table 3). Trichome density on the adaxial leaf surface of the GR Piketberg biotype was significantly higher compared to the GS Fauresmith biotype but did not significantly differ between the GR Swellendam and the GS George biotype (Table 2). The trichome densities on the adaxial surface were highest in the glyphosate-resistant Piketberg biotype, at 7.66 trichomes per mm^2^, followed by the glyphosate-resistant Swellendam biotype—6.68 trichomes per mm^2^, glyphosate-susceptible George biotype—6.41 trichomes per mm^2^, and glyphosate-susceptible Fauresmith biotype—5.06 trichomes per mm^2^ (Table 2). 

#### 2.3.3. Quantification of Leaf Wax Mass by Extraction in Chloroform

Data analysis showed non-significant differences in the amount of wax per unit leaf fresh weight extracted from leaf surfaces (*p* = 0.3533) (Table 4). The mean wax mass represents total waxes extracted from adaxial and abaxial leaf surfaces. 

### 2.4. Inheritance Studies

#### 2.4.1. Whole Plant Glyphosate Dose–Response 

A dose–response study was carried out on the parental lines and F2 generation. There was progressive chlorosis and eventual death in GS plants while GR plants produced new leaves and the pigmentation changed from yellow to green (Figure 4). The GR_50_ of the resistant Swellendam parental line was found to be 121.2-fold more resistant than the susceptible George parental line, whose GR_50_ value was 7.81 g ae ha^−1^ (Table 5).

#### 2.4.2. Segregation in F2 Generation

Second-generation (F2) plants were sprayed with glyphosate at the recommended label rate of 900 g ae ha^−1^ to discriminate between susceptible and resistant phenotypes. In response to glyphosate application, the chi-square tests for goodness of fit showed that the observed segregation ratios were in accordance with the expected frequencies for a 3:1 (R:S) single model (χ^2^ = 3.841, *p* < 0.05) (Table 6). The GR_50_ values for the F2 generation were intermediate to those of the R and S parents (Table 5) but closer to the resistant parent than susceptible parent. The S/R and R/S F2 generations were 46- and 88-fold more resistant than the susceptible parent, respectively (Table 5). 

## 3. Discussion 

Glyphosate-susceptible (GS) biotypes showed more herbicidal phytotoxic effects than glyphosate-resistant (GR) biotypes in dose–response experiments. In all biotypes, young developing leaves at the shoot apexes became chlorotic, and in GS biotypes, the chlorosis progressed into necrotic lesions on older leaves, with plants eventually dying at 21 days after spraying. In GR plants that survived, even at the highest glyphosate dose, there was regeneration of new leaves at the center of rosettes. This observation was similar to that reported in other studies on the response of *C. bonariensis* to glyphosate [35,36] and *C. canadensis* [37].

Resistance factors of biotypes ranged from 2.7- to 24.8-fold compared to the most susceptible biotype, with the highest resistance factors being from biotypes whose seed was collected from vineyards which had a long history of regular glyphosate use. Similar resistance factors have been documented by Travlos and Chachalis [38], who showed resistance factors ranging from four- to seven-fold in *C. bonariensis* growing in an orchard and vineyards, while Urbano et al. [39] registered 7- to 10-fold resistance factors in *C. bonariensis* from olive groves. Comparable resistance factors have been reported in other glyphosate-resistant weeds including *C. canadensis* [40], ryegrass [41], Palmer amaranth (*Amaranthus palmeri* S. Wats.) [42] and *E. colona* [43]. In the present study, the varying resistance factors in biotypes is an indication that *C. bonariensis* may still be segregating into populations that are either glyphosate-resistant or -susceptible, depending on the level of selection pressure. In general, higher resistance factors were demonstrated in biotypes collected from vineyards, where the cropping system involves repeated use of glyphosate, and in the case of conservation tillage systems that are dependent on glyphosate use, as well as in systems lacking crop and/or herbicide rotation. 

While target-site mutations that have been associated with two- to five-fold glyphosate resistance have been reported in other weed species [16], these mutations were not detected in the glyphosate-resistant biotypes examined. No mutations coding for amino acid positions Pro106, Thr102 and Ala103 were found in the *EPSPS* gene sequence. This indicates that a glyphosate resistance mechanism based on a target-site mutation in the *C. bonariensis* biotypes tested is unlikely. Similar observations were reported in *C. bonariensis* [44,45,46], *C. canadensis* [47,48], giant ragweed (*Ambrosia trifida* L.) [49], annual bluegrass (*Poa annua* L.) [50], windmill grass (*Chloris truncata* R. Br.) [51] and *A. palmeri* [52]. 

Results from the quantitative RT-PCR on cDNA showed up to two-fold higher *EPSPS* mRNA levels compared to the untreated control in the susceptible Fauresmith biotype (Figure 3). The transcript levels in glyphosate-resistant biotypes were comparable or lower than those recorded for glyphosate-susceptible Fauresmith and George biotypes. It is interesting to note that the highest expression level was registered in the susceptible biotype from Fauresmith. These results clearly indicate that the response of the resistant biotypes tested in this study is not due to *EPSPS* overexpression. Similar findings were reported in *C. bonariensis* [49], *C. canadensis* [47], *A. tuberculatus* [14], *E. colona* [7], *L. multiflorum* [53], *A. palmeri* [54], *E. indica* [12] and greater beggarticks (*Bidens subalternans* DC.) [55]. These results differ from previous reports of *EPSPS* overexpression as the mechanism of glyphosate resistance in *C. bonariensis* from Spain [19], *C. canadensis* [37], *A. palmeri* [52], ripgut brome (*Bromus diandrus* Roth) [56], *E. indica* [57] and *C. truncata* [51]. If we assume that the high resistance factors reported in the tested GR biotypes are attributed to a highly effective mechanism of resistance, then a non-target-site mechanism is most likely to provide the enhanced levels of glyphosate resistance in the studied accessions [15,41,58].

The cuticle thickness was not significantly different in biotypes from Piketberg (GR) and Fauresmith (GS), although in the GR Swellendam biotype, the cuticle was significantly thicker than in both the GS Fauresmith and GS George biotypes. The average cuticular membrane thickness ranged between 0.11 µm in Swellendam (GR) and 0.07 µm in George (GS) biotypes. Cuticle thickness has been documented to vary between 0.1 and 10 µm [59]. Heredia [60] reported cuticle thickness of 0.35 µm in a grape vine. The cuticle thickness reported in this study is close to the documented cuticle thickness, though slight differences could be due to variability in plant species. Given that there were insufficient mean separations in cuticle thickness between the GR and GS biotypes, the differences in glyphosate response in biotypes is unlikely to be related to the cuticle thickness. Ultrastructural images of cuticles of all investigated biotypes were highly corrugated or folded (Appendix A). Cuticular folding is a feature common in many plants and has been reported in ivyleaf morningglory (*Ipomoea hederacea* L.), pitted morningglory (*Ipomoea lacunose* L.) and palmleaf morningglory (*Ipomoea wrightii* Gray) [61] and false turkey-berry (*Plectroniella armata* K.Schum.) [62]. According to Koch et al. [63], cuticular folding, which forms part of the cell surface structuring, could result from subcuticular inserts of mineral crystals such as silicon oxide or natural folding of the cuticle itself. The function of cuticular folds has been linked to an increased surface area for the facilitation of the active exchange of substances or metabolites [62].

*Conyza bonariensis* is also called ‘*hairy fleabane*’ because of the presence of dense trichomes on the leaf surfaces, a phenomenon believed to contribute to unsatisfactory control of the weed by foliar-applied herbicides [64]. More trichomes were found on the adaxial as opposed to abaxial surfaces. Comparable results were reported by Wu and Zhu [64] and Procopio et al. [65] in *C. bonariensis* and in wild Indian mustard (*Brassica juncea* L.) [66]. However, Burrows et al. [67] reported contrary results, with the trichome numbers being higher on the abaxial than adaxial leaf surfaces in silverleaf nightshade (*Solanum elaeagnifolium* Cav.). 

Trichome density on the adaxial leaf surface was significantly higher in the GR Piketberg biotype compared to the GS Fauresmith biotpe, but not significantly different in the case of GR Swellendam and GS George biotypes (Table 2). Trichome densities on adaxial leaf surface ranged from 5.06 to 7.66 trichomes per mm^2^. In related investigations, much higher trichome densities have been reported. Wu and Zhu [64] established 67.2 to 221.9 trichomes per mm^2^ while Procopio et al. [65] reported 35.4 trichomes per mm^2^ in *C. bonariensis*. In other weed species, Burrows et al. [67] demonstrated trichome densities of 35 trichomes per mm^2^ in *S. elaeagnifolium* and Huangfu et al. [66] established up to 49.6 trichomes per mm^2^ in wild *B. juncea*. This study has shown that the response of biotypes to glyphosate cannot be attributed to trichome density. 

Epicuticular waxes have been documented to form a primary barrier to spread, penetration and absorption of water-soluble foliar-applied herbicides because of their hydrophobic nature [59]. In the present study, there were no significant differences in total leaf wax among the four *C. bonariensis* biotypes. The wax per unit leaf fresh weight varied from 2107.7 to 2984.7 µg g^−1^. Similar results have been reported in other weed species. Koger and Reddy [68] reported non-significant differences between resistant and susceptible biotypes of *C. canadensis*, regardless of their origin. The amount of wax per fresh weight ranged from 191 to 3097 µg g^−1^. Nandula et al. [69], working on *L. multiflorum* from Mississippi, also reported non-significant differences among tested biotypes, with wax amount varying from 1314 to 1413 µg g^−1^ fresh weight. Therefore, results from our research have demonstrated that differences in tolerance of *C. bonariensis* biotypes to glyphosate doses cannot be explained conclusively by the amount of epicuticular waxes on leaves. 

The lower resistance to glyphosate in the second generation (F2) plants compared with the GR parent (Swellendam) suggested that resistance is inherited in an incompletely dominant manner at the glyphosate application rate of 900 g ae ha^−1^. The GR_50_ values for the F2 generation were 359.25 g ae ha^−1^ (S/R) and 687.64 g ae ha^−1^ (R/S), implying that the level of glyphosate resistance is intermediate between the parental individuals. Comparable results were reported in *C. canadensis* [29]. According to Murray et al. [70], had the dose–response curves resembled resistant or susceptible parental lines, then the inheritance pattern would have been fully dominant or recessive, respectively. An incompletely dominant type of glyphosate resistance inheritance has been previously reported in *C. canadensis* [30], *E. indica* [71] and *L. rigidum* [31,32,72]. The practical implication of the inheritance of glyphosate resistance established in this study is that resistance is most likely to spread rapidly in the field with repeated applications of glyphosate herbicide. This is attributed to the fact that 75% of the progeny from a cross of heterozygous individuals will exhibit the glyphosate resistance phenotype [32]. In addition, the exceptionally prolific nature of *C. bonariensis* in producing more than 100,000 seeds per plant [73] and dispersal by wind over distances of 100 km [74] would increase the spread of resistant individuals to adjacent districts.

## 4. Materials and Methods 

### 4.1. Plant Material Sources

The six *C. bonariensis* populations used in this study were previously described by Okumu et al. [35]. The populations were selected based on distinctive response to glyphosate treatment, as either being resistant (GR) or susceptible (GS), and showing high seed viability, germination and establishment. These biotypes included: two GS biotypes from George and Fauresmith with resistance factors of 1.0- and 1.3-fold, respectively, and four GR biotypes from Durbanville, Worcester, Swellendam, and Piketberg, with resistance factors (RF values) of 3.0-, 9.5-, 20.4-, and 26.9-fold, respectively [35]. Three biotypes were collected from vineyards, one from a wheat field, one from a hops orchard and one from a pasture field (Table 7). For the leaf morphology experiments, four biotypes were used: George (GS), Fauresmith (GS), Swellendam (GR) and Piketberg (GR). 

### 4.2. Glyphosate Dose–Response

This study was carried out according to the method described by Beckie et al. [75] and Okumu et al. [35]. Seeds were grown in pots and kept in a glasshouse at Hatfield experimental farm at the University of Pretoria. Plants were sprayed with Roundup^®^ Turbo herbicide (Monsanto South Africa (Pty) Ltd., Sandton, South Africa) when they were at the rosette stage, with 4 to 6 fully unfolded leaves (6–8 cm rosette diameter). The glyphosate herbicide rates used were: 0 (untreated control), 225, 450, 900, 1800 and 3600 g ae ha^−1^. The registered label rate in South Africa is 900 g ae ha^−1^. All of the plants were clipped at the soil surface 21 days following treatment. Only green foliage was harvested, weighed and recorded as fresh weight. Zero biomass accumulation was recorded for sensitive plants that died (no green parts visible). The trial was arranged in a completely randomized design with five replicates per herbicide dose per biotype (6 biotypes × 6 doses × 5 replicates). Each pot represented a replicate. A three-parameter non-linear regression model, described by Seefeldt et al. [76], was used to generate dose–response curves in R version 3.2.3 (R Development Core Team, Vienna, Austria) using the ‘drc’ package [77]. Dose–response curves were used to ascertain the herbicide dose that causes a mean fresh weight reduction or injury of 50% (GR_50_) in *C. bonariensis*. The above-ground fresh weight was expressed as a percentage of the mean untreated control. The resistance factor (RF) was determined by dividing the GR_50_ of a population by the GR_50_ of the most sensitive population (George). 

### 4.3. Molecular Characterization of EPSPS Gene 

#### 4.3.1. Partial Sequencing of *EPSPS* cDNA

##### Plant Material, RNA Extraction and cDNA Synthesis

Glyphosate-resistant (GR) plants from Swellendam, Piketberg and Worcester and glyphosate-susceptible (GS) plants from George and Fauresmith were used in the study. Plants were grown in a pot experiment in the glasshouse and at the 4–6 leaf stage (rosette), leaf samples were harvested, frozen in liquid nitrogen and stored at −80 °C until use. Samples were ground in liquid nitrogen using a pestle and mortar. Total RNA was extracted from 100 mg of the sample powder using TRIzol reagent (Invitrogen, Waltham, MA, USA) according to the manufacturer’s protocol. RNA purification and *DNase* I digestion was carried out using Direct-zol RNA MiniPrep (Zymo Research, Irvine, CA, USA) following the manufacturer’s instructions. The RNA quantity and purity were determined spectrophotometrically using a Nanodrop^R^ (Thermo Scientific, Waltham, MA, USA) and by gel electrophoresis run on a 1% (*w*/*v*) agarose gel in 1× Tris-acetate-EDTA (TAE) buffer, stained with GelRed^TM^ (Biotium, Freemont, CA, USA) added to 6× Loading Dye (Thermo Scientific, Waltham, MA, USA), and the whole set up run at 100 V for 20 min.

Extracted RNA was used to synthesize single-stranded cDNA by random hexamer priming with the GoScript^TM^ Reverse Transcription System kit (Promega, Madison, WI, USA) in a 38 μL reaction mixture, according to the manufacturer’s protocol. Polymerase chain reaction (PCR) was conducted to amplify a 492 bp section of the *EPSP synthase* gene containing proline codon at position 106 where resistance-conferring point mutation(s) have been reported. The forward (5′-CTGTCTGAGGGGACTACT-3′) and reverse (5′-TATCTCTACGTCTCCCAG-3′) primers were designed based on the *C. bonariensis EPSP synthase* mRNA sequence, GenBank accession number EF200069.1. The PCR was performed in a 10 μL reaction mixture containing 0.5 μL of primers (10 μM), 5 μL One*Taq* 2x Master Mix with Standard Buffer (New England Biolabs, Ipswich, MA, USA), 1 μL of the synthesized cDNA and 3 μL double-distilled water (ddH_2_O). The PCR reactions were performed in a thermal cycler (Bio-Rad, Hercules, CA, USA) using the following PCR program: initial denaturation at 94 °C for 5 min, 39 cycles of 94 °C for 30 s, annealing at 58 °C for 30 s, extension at 72 °C for 1 min and a final elongation at 72 °C for 10 min. An aliquot of the PCR product was loaded on a 2% (*w*/*v*) agarose gel in 1× Tris-acetate-EDTA (TAE) buffer, and stained with GelRed^TM^ (Biotium, Freemont, CA, USA) added to 6× Loading Dye (Thermo Scientific, Waltham, MA, USA) to ensure that the correct band was amplified. The quantity and purity of cDNA was further verified using a Nanodrop^R^ Spectrophotometer (Thermo Scientific, Waltham, MA, USA). The rest of the PCR product was purified using ExoSAP-IT^TM^ PCR Product Cleanup (Thermo Scientific, Waltham, MA, USA) to remove excess primers and nucleotides, according to the manufacturer’s protocol. The PCR products were sequenced in both directions using the BigDye^TM^ Terminator v3.1 Cycle Sequencing Kit (Thermo Scientific, Waltham, MA, USA) following the manufacturer’s instructions. The sequenced products were precipitated in an ethanol and sodium acetate wash and resuspended in 10 μL of formamide before being sent to the University of Pretoria Sequencing Facility for sequencing. PCR products of the five *C. bonariensis* biotypes were sequenced in both forward and reverse directions to minimize sequencing errors. The DNA sequence chromatograms were visually checked for quality and consistency before they were assembled, compared and analysed using the ContigExpress from the Vector-NTi Advance 11.5 programs (Invitrogen, Carlsbad, CA, USA). The sequences were aligned to the *EPSP synthase* sequence of *C. bonariensis* from the GenBank (Accession number EF200069.1) as a reference. The aim was to determine the presence of mutation(s) as a possible target-site mechanism of glyphosate resistance in GR biotypes.

#### 4.3.2. RNA Expression

##### Plant Establishment and Glyphosate Treatment

Four biotypes, Swellendam (GR), Piketberg (GR), George (GS) and Fauresmith (GS), were established in the glasshouse at the Hatfield experimental farm. At the 4–6 leaf stage (rosette), plants were treated with glyphosate, Roundup^®^ Turbo, (Monsanto South Africa (Pty) Ltd., Sandton, South Africa), at the registered label recommended rate of 900 g ae ha^−1^. Control plants were not treated with the herbicide. Aboveground plant tissue was sampled at 0 (before treatment), 4 and 6 days after treatment (DAT). Leaf samples were harvested, frozen in liquid nitrogen and stored at −80 °C until use.

##### RNA Extraction and cDNA Synthesis 

Grinding of samples in liquid nitrogen followed by RNA extraction and cDNA synthesis was carried out as described in the DNA sequencing experiment. The RNA and cDNA quantity and purity was determined spectrophotometrically using a Nanodrop^R^ (Thermo Scientific, Waltham, MA, USA) and by gel electrophoresis, as described before. 

##### Primer Design

Primers for *EPSP synthase* gene were designed based on the published sequence of *Conyza canadensis*, GenBank accession number AY545667. *Actin* gene was used as a reference gene, and the primers were constructed based on *Chrysanthemum seticuspe* f. *boreale* mRNA sequence, GenBank accession number AB770470.1. Primer3 software version 2.3.7 (Primer3web, Tartu, Estonia, http://primer3.ut.ee/ (accessed on 27 October 2016)) was used in the design of all the primers. Primers were selected using the following criteria: primer length between 18 and 25 bases, amplicon length between 80 and 150 bp and annealing temperature (Tm) between 55 and 65 °C, with the difference in Tm between respective forward and reverse primers being less than 4 °C (Appendix A). 

##### Quantitative Reverse-Transcription Polymerase Chain Reaction (qRT-PCR) Analysis

Primer specificities were tested by performing a standard PCR on the cDNA using the following thermal cycler conditions: initial denaturation at 94 °C for 5 min, 39 cycles of 94 °C for 30 s, annealing at 61 °C for 30 s, extension at 72 °C for 1 min and a final elongation at 72 °C for 10 min. The PCR products were run on a 2% agarose gel. Quantitative RT-PCR was used to measure cDNA expression level of *EPSP synthase* gene relative to the *Actin* gene. Standard curves for the primers were performed using a 2x dilution series, ranging from 200 ng to 6.25 ng. The slope of the standard curve was used to determine amplification efficiency (E). Quantitative RT-PCR was conducted in a 10 μL reaction containing 5 μL iTaq^TM^ Universal SYBR Green Supermix (Bio-Rad, Hercules, CA, USA), 0.5 μL of each forward and reverse primers (10 μM), 2 μL double-distilled water (ddH_2_O) and 2 μL of cDNA. All qRT-PCR reactions were performed using a CFX96^TM^ Real-Time PCR System (Bio-Rad, Hercules, CA, USA). The initial denaturation step was held at 95 °C for 5 min, followed by 35 cycles of denaturation at 95 °C for 10 s, and a combined annealing/extension step at 60 °C for 30 s. Melt curve analysis was performed by heating the PCR products from 60 to 95 °C, in 0.5 °C increments for 5 s and reading the fluorescence at each step. The melt curve analysis was carried out to verify the primer specificity and that no primer dimers formed. Negative controls, which were carried out in triplicates, consisted of primers with no templates. Additionally, no reverse-transcriptase controls were included to validate the effectiveness of the *DNase* I digestion step. Melt curve analysis showed an absence of primer dimer formation and amplification in either primer set (Appendix A). Primer efficiencies were 98.98% for *EPSPS* and 102.97% for *Actin* genes (Appendix A). Data was analysed using CFX Manager Version 2.1.1022.0523 (Bio-Rad, Hercules, CA, USA). Relative quantification of *EPSPS* gene for each sample was calculated using the equation 2^−∆∆Ct^, where ΔCt = (Ct, *EPSPS*-Ct, *Actin*), according to a method described by [78,79]. The results obtained were expressed as the fold increase or decrease of *EPSPS* gene relative to *Actin* gene in treated samples compared with the untreated controls. The experiment was arranged in a completely randomized design with four biotypes, three biological replicates and three technical replicates. 

### 4.4. Leaf Morphological Characterization 

#### 4.4.1. Cuticle Thickness Examination 

##### Preparation of Plant Materials for Transmission Electron Microscopy (TEM)

Plants were established in the glasshouse as in the dose–response experiment. At the rosette stage (6–8 cm diameter with 4 to 6 leaves), one mature and fully expanded leaf from each of 10 plants per population was sampled and prepared for TEM examination. Each leaf represented a replicate, making a total of 10 replicates per biotype. Too-young and too-old leaves were avoided. Leaf segments of 1 mm^2^ in size from the middle of the leaf between the main vein and margin were excised in 2.5% glutaraldehyde. The leaf segments were further fixed in 2.5% glutaraldehyde in 0.075 M sodium phosphate buffer (pH 7.4) for two hours at room temperature and rinsed three times, 10 min each time in 0.075 M sodium phosphate buffer. The segments were then fixed in 0.5% aqueous osmium tetraoxide in a fume hood, followed by rinsing three times in distilled water and dehydrated in a series of ethanol concentrations: 30%, 50%, 70%, 90%, 100%, 100% and 100% for 10 min in each concentration. Infiltration of the segments was performed for one hour, sequentially in 30% and 60% epoxy resin and a further 4 h in 100% according to protocol specifications [80]. Segments were embedded in molds and polymerized at 60 °C for 39 h. Ultra-thin sections were made with a Reichert Jung Ultracut E microtone using a diamond knife, mounted on copper grids and contrasted in 4% aqueous uranyl acetate for 10 min. The sections were rinsed in distilled water and contrasted a second time in Reynolds’ lead citrate for 2 min. Sections were then mounted in a Phillips CM10 transmission electron microscope (TEM) operated at 80 kV for viewing and digital photographs of the plant cuticle taken at a magnification of x 64,000. The Olympus Soft Imaging System (SIS) iTEM software package was used to measure the cuticle thickness. The experiment was arranged in a completely randomized design. Data were subjected to analysis of variance using the PROC GLM procedure in SAS software version 9.3 (SAS 2002–2010, SAS Institute Inc., Cary, NC, USA) and mean comparison with Tukey’s test, *p* ≤ 0.05, to determine significant differences.

#### 4.4.2. Trichome Density 

Plants were planted in pots, and at the rosette stage, one mature and fully expanded leaf each from 10 plants per population were sampled for examination. One leaf represented one replicate, and a total of 10 replicates were used per biotype. For light microscopy (LM) work, whole leaves were viewed at ×40 objective and 10× ocular using a Zeiss SteREO Discovery.V20 microscope equipped with an AxioCam MRc5 camera. Digital photographs were taken and trichomes counted. The number of trichomes per mm^2^ on the adaxial and abaxial surfaces was recorded. The experiment was arranged in a completely randomized design. Data were subjected to analysis of variance using PROC GLM procedure in SAS software version 9.3 (SAS 2002–2010, SAS Institute Inc., Cary, NC, USA) and mean comparison with Tukey’s test, *p* ≤ 0.05, to determine significant differences. 

For scanning electron microscopy (SEM), leaf segments of 2 mm^2^ in size were prepared as in the TEM experiment until the final dehydration step. The segments were critical point dried in liquid carbon dioxide for 4 h, mounted on stubs using a two-way stick tape and splutter-coated with gold to reduce specimen charging. Viewing was performed using an Ultra Plus Zeiss Gemini scanning electron microscope (Carl Zeiss NTS gmbH, Oberkochen, Germany) and digital photographs taken to describe the characteristics of trichomes. 

#### 4.4.3. Quantification of Leaf Wax Mass 

Plants were grown in pots, and at the rosette stage, 50 mature and fully expanded leaves including the petiole were sampled per biotype. Of the 50 leaves sampled, one replicate and five replicates were used in total per biotype. The total fresh weight was recorded and wax extracted using a procedure previously presented by Sanyal et al. [81]. For each biotype and replicate, 50 leaves were immersed in 100 mL (grade) chloroform in a glass beaker for 20 s at room temperature in an ultra-sonic bath. The chloroform–wax solution was filtered using a syringe filter fitted with Durapore membrane filters (0.22 mm GV), and the volume reduced to approximately 20 mL in a rotary evaporator. The reduced chloroform–wax solution was transferred to a pre-weighed 25 mL glass scintillation vial. Chloroform was evaporated to dryness under a fume hood and further dried in a forced-air oven at 40 °C for 72 h to ensure complete dryness of the samples. The amount of wax was expressed as the wax mass per unit leaf fresh weight. The experiment was arranged in a completely randomized design. Data were subjected to analysis of variance using the PROC GLM procedure in SAS software version 9.3 (SAS 2002–2010, SAS Institute Inc., Cary, NC, USA) and mean comparison with Tukey’s test, *p* ≤ 0.05, to determine significant differences.

### 4.5. Inheritance Studies 

#### 4.5.1. Glyphosate Dose–Response for Identification of Parental Lines

Glyphosate-resistant (GR) and -susceptible (GS) parental lines used were derived from seeds collected from Swellendam and George localities in South Africa, respectively. Previous screening studies established GR_50_ values of 2958.62 g ae ha^−1^ for the resistant Swellendam biotype and 145.36 g ae ha^−1^ for the susceptible George biotype [35]. The GR and GS populations were established in pots in a glasshouse and selfed for two generations by placing clear plastic pollination bags over the capitula before anthesis. Clear pollination bags were used because they allow light and air to reach the plants whilst preventing insects from pollinating the flowers, thus enabling self-pollination to take place. Bags were left on until seed set, and seeds were harvested and bulked separately for each biotype. In each cycle of selfing, the resistant parent was treated with the label recommended rate of 900 g ae ha^−1^ glyphosate Roundup^®^ Turbo (Monsanto South Africa (Pty) Ltd., Sandton, South Africa) at the rosette stage. The susceptible parent was not sprayed due to its high sensitivity to glyphosate treatment. Although all efforts were made to ensure that the resistant parent was homozygous for glyphosate resistance by conducting two rounds of selection, the possibility that there were some heterozygous individuals in the subsequent crosses cannot be ruled out. 

#### 4.5.2. Controlled Reciprocal Crosses and Generation of F2 Progeny 

The inflorescence of *C. bonariensis* consists of white pistillate ray florets (female) on the periphery and yellow hermaphrodite disk florets in the capitulum center [82]. Before anthesis, capitula of plants serving as female parents were emasculated by removing disk florets using forceps under a magnifying lens. To restrict self-pollination, the remaining non-emasculated capitula were removed from plants. Emasculated capitula and capitula of plants acting as male parents were covered with clear pollination bags. The female flowers were pollinated two to three days after emasculation by rubbing them with pollen from the male parent. Pollination bags were immediately put back after pollination and the plants allowed to set seed. Male capitula were disposed of and seeds harvested when the pappus became visible. Reciprocal crosses were made in both directions to obtain F1 generation. In the reciprocal crosses, the first letter denotes the female parent, that is, female S x male R (S/R) and female R x male S (R/S). Sixty F1 seeds from each of the reciprocal crosses were grown in the glasshouse under the same conditions as the parents and selfed to produce F2 seeds; these seeds were used for the inheritance studies. 

#### 4.5.3. Parents and F2 Dose–Response Experiments

A whole plant dose–response experiment was carried out using glyphosate rates of 0 (untreated control), 225, 450, 900, 1800 and 3600 g ae ha^−1^ as described previously in the dose–response experiment. Twenty-one days after treatment (DAT), all the plants were clipped at the soil surface and only green shoot tissue was weighed and fresh weight recorded. A plant count was performed at 21 days after treatment (DAT). Typical glyphosate toxicity symptoms on *C. bonariensis* include yellowing of growth points that progress into necrotic lesions on leaves, wilting, and plant death. Visual examination was performed and plants that showed these symptoms were scored ‘dead’ while those that maintained the green pigmentation and regeneration of new shoots were scored ‘alive’. The observed segregation data in F2 generation were tested for goodness of fit to expected segregation ratios (3:1) using a chi-square test (α = 0.01) [83]. GR_50_ values were estimated from a three-parameter non-linear regression model applied to the mean above-ground fresh weight, as previously described in the dose–response experiment. 

## 5. Conclusions 

The *C. bonariensis* biotypes tested in this study varied in response to glyphosate, with a wide range of response levels between glyphosate-susceptible (WP 19-George and WP 42-Fauresmith) and glyphosate-resistant biotypes (WP 22-Swellendam, WP 28-Worcester, WP 33-Durbanville and WP 38-Piketberg). The resistance factors (RF) for the biotypes were 1.0-fold for George, 2.7-fold for Fauresmith, 4.7-fold for Durbanville, 8.1-fold for Worcester, 13.6-fold for Swellendam, and 24.8-fold for Piketberg.

The present study used molecular approaches to establish the mechanism(s) of glyphosate resistance in *C. bonariensis*, and the mode of glyphosate resistance inheritance. Although glyphosate resistance is inherited in an incompletely dominant manner, target-site mutations and *EPSPS* mRNA overexpression are unlikely to be the mechanisms of glyphosate resistance in the tested biotypes. Impaired translocation was indicated as one of the mechanisms of glyphosate resistance in *C. bonariensis* from Spain [19]. Given the high glyphosate resistance factors established in the resistant biotypes, it is feasible that non-target mechanisms of resistance, including impaired translocation and vacuolar sequestration, could be responsible. Our previous study showed that glyphosate resistance in *C. bonariensis* is influenced by temperature [35]. Past investigations have determined that vacuolar sequestration and the overexpression of ABC-transporter genes (*M10* and *M11*) are temperature dependent [21,84]. We recommend further research in these areas as possible mechanisms of glyphosate resistance in *C. bonariensis* in South African biotypes.

The influence of leaf morphological characteristics of *C. bonariensis* on plant growth responses to glyphosate did not produce conclusive results. Differences in the response of biotypes to glyphosate treatment cannot be explained by differences in the leaf morphological characteristics investigated because of non-significant results in the amount of leaf wax and insufficient mean separations in cuticle thickness and trichome density data. The hypothesis that differences in response to glyphosate treatment could be attributed to leaf morphological characteristics was rejected.

## Figures and Tables

**Figure 1 plants-11-02830-f001:**
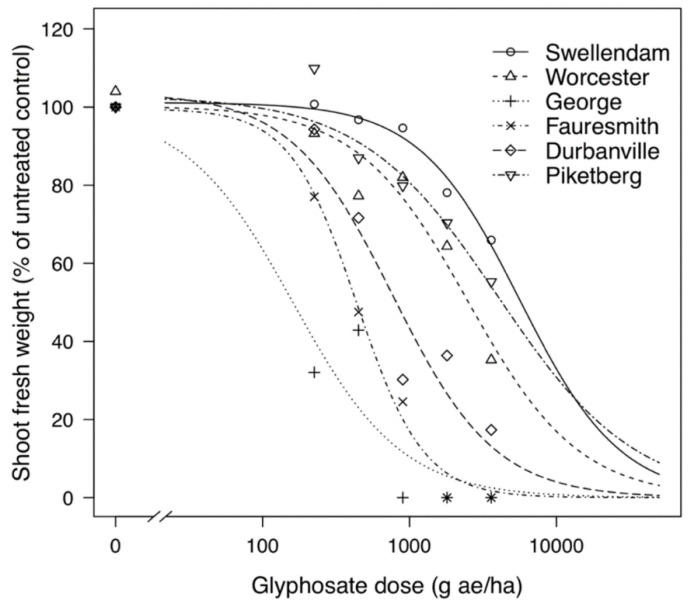
Growth responses of glyphosate-resistant (Swellendam, Worcester, Durbanville and Piketberg) and glyphosate-susceptible (George and Fauresmith) *C. bonariensis* biotypes to a range of glyphosate doses. The predicted curves are described by a three-parameter equation. (*) represents zero biomass accumulation.

**Figure 2 plants-11-02830-f002:**
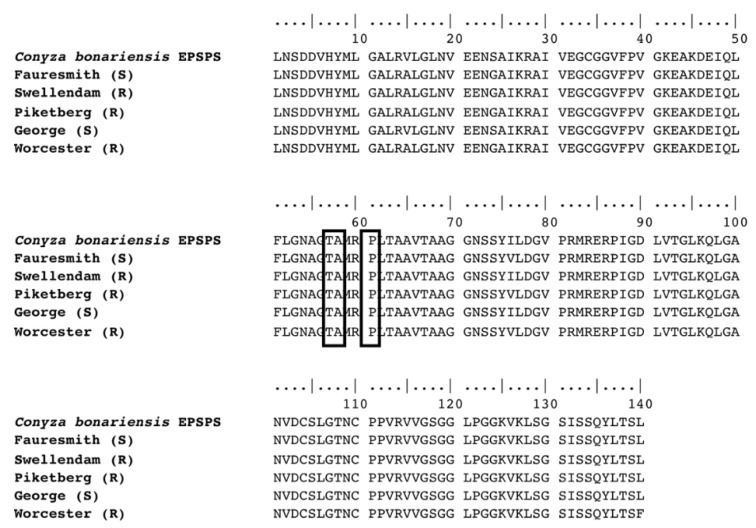
Amino acid sequence alignment of *EPSPS* cDNA isolated from GR (Swellendam, Piketberg and Worcester) and GS (Fauresmith and George) *C. bonariensis* plants. The boxed codons indicate a lack of mutations at the *EPSPS* Thr102, Ala103 and Pro106 positions.

**Figure 3 plants-11-02830-f003:**
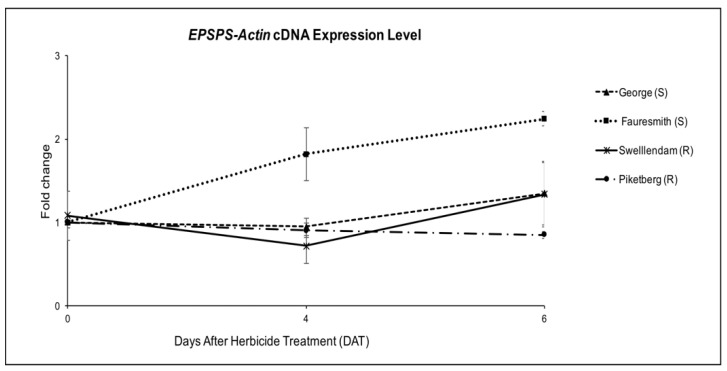
*EPSPS* gene expression relative to *Actin* gene for *C. bonariensis* biotypes at four and six days after treatment (DAT) with glyphosate at 900 g ae ha^−1^. Quantitative PCR was calculated using 2^−∆∆Ct^, where ΔCt = (Ct, *EPSPS*-Ct, *Actin*). Vertical bars represent the mean ± standard error.

**Figure 4 plants-11-02830-f004:**
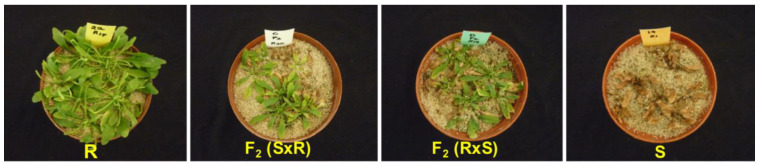
Response of *C. bonariensis* populations to glyphosate treatment at 900 g ae ha^−1^, 21 DAT. ‘R’ and ‘S’ represent the resistant (Swellendam) and susceptible (George) parental lines, respectively. S/R refers to female S × male R cross and R/S denotes female R × male S cross.

**Table 1 plants-11-02830-t001:** GR_50_ estimates from the dose–response regression analysis for *C. bonariensis* biotypes whose seeds were collected from five locations in the western and southern Cape regions, and one (Fauresmith) in the Free State province of South Africa.

Population Code	Location	Habitat	GR_50_ ^a^ (g ae ha^−1^)	RF ^b^
WP 19	George	Orchard	161.99 (65.19)	1.0
WP 42	Fauresmith	Pasture	437.31 (81.39)	2.7
WP 33	Durbanville	Vineyard	768.82 (170.86)	4.7
WP 28	Worcester	Vineyard	1314.80 (222.64)	8.1
WP 22	Swellendam	Wheat	2203.31 (646.58)	13.6
WP 38	Piketberg	Vineyard	4014.79 (1710.32)	24.8

^a^ Abbreviations: GR_50_—glyphosate dose required to result in 50% decrease in aboveground shoot fresh weight in *C. bonariensis*, 21 days after treatment (DAT). ^b^ Resistance factor calculated by the ratio of GR_50_ value of an individual *C. bonariensis* biotype to the GR_50_ value of the most susceptible biotype (George).

**Table 2 plants-11-02830-t002:** Cuticle thickness and trichome density of four biotypes of *C. bonariensis* plants established in the glasshouse.

		Leaf Morphological Measurements	
Biotype Code	Location	Cuticular Membrane Thickness in µm ^a^	Trichome Density on Adaxial Leaf Surface per mm^2 a^
WP 22	Swellendam	0.11 (0.004) ^a^	6.68 (0.51) ^ab^
WP 38	Piketberg	0.09 (0.005) ^ab^	7.66 (0.38) ^a^
WP 42	Fauresmith	0.08 (0.005) ^bc^	5.06 (0.44) ^b^
WP 19	George	0.07 (0.006) ^c^	6.41 (0.37) ^ab^

Mean cuticular membrane thickness and trichome density measurements ± standard error. ^a^ Means within a column followed by the same letter are not significantly different according to Tukey’s test, *p* ≤ 0.05.

**Table 3 plants-11-02830-t003:** Mean trichome density (average of 10 replicates) of adaxial and abaxial leaf surfaces of four biotypes of *C. bonariensis* plants.

Surface	Trichome Density on Leaf Surface per mm^2 a^
Adaxial	6.52 (0.42) ^a^
Abaxial	5.79 (0.51) ^b^

Trichome density ± standard error. ^a^ Means within a column followed by the same letter are not significantly different according to Tukey’s test, *p* ≤ 0.05.

**Table 4 plants-11-02830-t004:** Mean leaf wax mass per unit fresh weight of four biotypes of *C.*
*bonariensis* plants.

Biotype Code	Location	Wax Mass per Unit Leaf Fresh Weight (µg g^−1^) ^a^
WP 38	Piketberg	2984.7 (157.6) ^a^
WP 42	Fauresmith	2396.2 (191.9) ^a^
WP 22	Swellendam	2370.8 (472.0) ^a^
WP 19	George	2107.7 (416.3) ^a^

Estimates represent mean leaf wax ± standard error. ^a^ Means within a column followed by the same letter are not significantly different according to Tukey’s test, *p* ≤ 0.05.

**Table 5 plants-11-02830-t005:** Estimates of dose–response analysis as calculated from 3-parameter log-logistic nonlinear regression model of glyphosate dose–response resulting in 50% growth reduction (GR_50_).

Population	GR_50_ (g ae ha^−1^) ^a^	RF ^b^
George (S)	7.81 (2.33)	1
Swellendam (R)	946.47 (226.99)	121.20
F2 (S × R)	359.25 (82.68)	46.0
F2 (R × S)	687.64 (164.89)	88.0

^a^ Abbreviations: GR_50_—glyphosate dose required to reduce aboveground shoot fresh weight by 50% in *C. bonariensis*, 21 days after treatment (DAT). ^b^ Resistance factor calculated by the ratio of GR_50_ value of respective *C. bonariensis* population/generation to the GR_50_ value of susceptible parent, George.

**Table 6 plants-11-02830-t006:** Segregation of glyphosate resistance phenotype in F2 generation of *C. bonariensis* 21 DAT with glyphosate at 900 g ae ha^−1^.

Population	Phenotype		Total	*df*	χ^2^ (3:1) ^b^
	Alive (R) ^a^	Dead (S) ^a^			
F2 (S × R)	105	37	142	1	0.09
F2 (R × S)	113	38	151	1	0.02
Pooled second generation (F2) plants	218	75	293	1	0.06

^a^ Phenotypes were visually scored for survival and classified as either Alive (R) or Dead (S). ^b^ Critical value (χ^2^) values were obtained from results of chi square tests for goodness of fit (α < 0.05) to a 3:1 (R:S) segregation model with χ^2^ = 3.841.

**Table 7 plants-11-02830-t007:** Geographic location and habitat of populations from which *C. bonariensis* seeds were sampled mainly in the western and southern Cape regions of South Africa.

Population Code	Location	Glyphosate Tolerance Status	GPS Coordinates	Habitat
WP 19	George	Susceptible	033°50.955′ S 022°21.328′ E	Hops Orchard
WP 42	Fauresmith *	Susceptible	-	Pasture
WP 33	Durbanville	Resistant	033°47.646′ S 018°40.226′ E	Vineyard
WP 28	Worcester	Resistant	033°35.713′ S 019°31.149′ E	Vineyard
WP 22	Swellendam	Resistant	034°07.451′ S 020°21.809′ E	Wheat
WP 38	Piketberg	Resistant	033°08.752′ S 019°00.341′ E	Vineyard

* Sampled on a land with alfalfa in Fauresmith district, Free State province.

## Data Availability

Not applicable.

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
