# Peer review of "The Molecular, Morphological and Genetic Characterization of Glyphosate Resistance in Conyza bonariensis from South Africa"

_plants, 2022, doi:10.3390/plants11212830_

Round 1

Reviewer 2 Report

Okumu et al. demonstrated leaf phenotypical difference of Glyphosate resistance biotype of hairy fleabane (in South Africa) compared to Glyphosate susceptible biotype. And they added observation of EPSPS gene variation in the same biotypes. The microscopic images and leaf morphological analysis look interesting observations to understand glyphosate resistance.  However, in my opinion, they need to turn their main focus and replace suitable figures for better understanding of glyphosate resistance. 

1)    Most of all, the favorable leaf morphology against glyphosate treatment in GR lines is the main concept of this manuscript. Therefore, they’d better to focus on most experiments to discuss aberrant components on leaf surface including molecular and physical evidence between GR vs GS. 

2)    They need the image to show different trichome distribution image between GR and GS. Figure 5 looks not necessary in this document. Also, I could not see any description about Figure 5 in the main body. We’d better to remove or move this figure to supplemental data.   

3)    In my opinion, the mutation and gene expression of EPSPS is not important concept in this document. They’d better to check the gene expression or variation of the leaf shape regulatory genes or trichome development-related genes between GR and GS. They can find many homologous candidate genes from Arabidopsis leaf developmental gene family. Also, if they include the new gene expression of leaf morphology-related genes, they need more citation in introduction.

4)    The explanation about EPSPS gene is not novel. What is the relationship of leaf morphology with this gene? Also, please consider the Figure 6 and 7 novelty or emphasize the new observation in their study if they hope to discuss EPSPS relating to current observation.

I also suggest minor changes in this manuscript. 

. In ln 115, they describe the correct abbreviation of R/S factors for reader.  

. Why don’t you make a summary table combining Table 2,3, and 4 to overview in the same concept? Then, discuss about leaf morphology from each biotype. 

Reviewer 3 Report

The manuscript entitled, "The Morphological, Molecular and Genetic Characterization of Glyphosate Resistance in Conyza bonariensis (L.) Cronquist from South Africa" is a well-written article and interesting one. However, there are some recommendations that must be considered before getting the paper accepted. 

1. Need to add the present research gap for what the study was designed.

2. What are the hypothesis?

3. Add this reference in line no. 76 https://doi.org/10.1016/j.indcrop.2019.111710

4. Mention the methodology clearly.

5. What is the final recommendation of this study?

Reviewer 4 Report

I recommend publishing the manuscript, entitled “The Morphological, Molecular and Genetic Characterization of Glyphosate Resistance in Conyza bonariensis from South Africa” in Plants journal with minor changes

Round 2

Reviewer 1 Report

L. 347, Ripgut brome in lowercase

L. 373, comma instead of semicolon

Reviewer 2 Report

Thank you for their revision. This version looks much better than the first submission. I agree on the publication of this version but still suggest the profiling of leaf morphology related gene expression in the future.

Also, as a minor suggestion, please make superscript the marker of statistical significancy in all tables.   Please double check most symbols (such as X for times). Please move Table 8 to supplementary materials. 
